# Electronic Cigarette Use and Smoking Initiation in Taiwan: Evidence from the First Prospective Study in Asia

**DOI:** 10.3390/ijerph16071145

**Published:** 2019-03-30

**Authors:** Yu-Ning Chien, Wayne Gao, Mattia Sanna, Ping-Ling Chen, Yi-Hua Chen, Stanton Glantz, Hung-Yi Chiou

**Affiliations:** 1Chung-Hua Institution for Economic Research, Taipei 106, Taiwan; chienyuning@cier.edu.tw; 2College of Public Health, Taipei Medical University, Taipei 110, Taiwan; yichen@tmu.edu.tw; 3Master’s Program in Global Health and Development, Taipei Medical University, Taipei 110, Taiwan; waynegao@tmu.edu.tw (W.G.); msanna@tmu.edu.tw (M.S.); 4Graduate Institute of Injury Prevention and Control, Taipei Medical University, Taipei 110, Taiwan; plchen@tmu.edu.tw; 5Department of Medicine, Center for Tobacco Control Research and Education, University of California San Francisco, San Francisco, CA 94117, USA; Stanton.Glantz@ucsf.edu

**Keywords:** e-cigarette, smoking initiation, prospective study, Asian youth, Taiwan

## Abstract

A growing literature indicates that electronic cigarette use increases the risk of subsequent initiation of conventional smoking among cigarette-naïve adolescents in several Western countries. This research assesses the same relationship in an Asian country, Taiwan. The Taiwan Adolescent to Adult Longitudinal Study is a school-based survey that was carried out in two waves in 2014 (baseline) and in 2016 (follow-up). It employs probability sampling to create nationally representative samples of students in junior high school (mean age 13, 7th grade at baseline) and in senior high school (mean age 16, 10th grade at baseline). Data from this survey were analyzed via logistic regression to estimate the association between ever use of e-cigarettes at baseline and smoking initiation at follow-up, accounting for susceptibility to smoking, socio-demographic profile, depression status, and peer support. Among the 12,954 cigarette-naïve students surveyed, those with e-cigarette experience at baseline exhibited higher odds of smoking initiation at follow-up (Odds Ratio = 2.14, 95% CI (1.66, 2.75), *p* < 0.001). For the first time, we confirmed, through a longitudinal survey, a prospective association between ever use of e-cigarettes and smoking initiation in an Asian adolescent population. The restrictive policy on e-cigarettes currently in force in Taiwan is justified to prevent both e-cigarette and cigarette use among adolescents.

## 1. Introduction

The popularity of electronic cigarettes among teenagers and adolescents is increasing in the U.S. [1] and other developed countries [2,3,4], raising concerns in the public health community and in many governments [5,6], particularly when youth who have never smoked cigarettes are involved. Longitudinal studies conducted in the U.S. [7,8,9,10,11,12,13,14], Canada [15,16], Germany [17], and the U.K. [18,19,20] have demonstrated that never cigarette smoking youth who have tried e-cigarettes are more likely to initiate at a later point [7,8,10,13,20,21,22,23,24] and that e-cigarette users are more inclined towards other substance use [25]. In accordance with these findings, the European Tobacco Products Directive stated: “*Electronic cigarettes can develop into a gateway to nicotine addiction and ultimately traditional tobacco consumption, as they mimic and normalize the action of smoking*” [26], while the Australian National Health and Medical Research Council concluded that actions should be taken to minimize the harm of e-cigarettes to users and bystanders and to protect vulnerable groups such as young people [27].

As of 2018, in Taiwan, e-cigarette liquid with nicotine is classified as an illegal pharmaceutical product and advertising nicotine-free e-cigarettes as smoking cessation devices is prohibited. The current regulation is seemingly strict, but in fact has loopholes and has raised unsolved controversies [4], with the result that vaping products are still readily available, especially over the Internet. As for age distribution, e-cigarettes are more widespread among youth (2.5% among junior high school students and 4.5% among senior high school students in 2017 [28]) than among adults (0.5% in 2017 [29]).

Even though e-cigarette use has been significantly increasing in many Asian countries, the effects on Asian youth have only been investigated through cross-sectional studies [3,30,31]. Thus, this research aimed to examine the impact of ever using e-cigarettes on smoking initiation among cigarette-naïve individuals in a longitudinal sample of Taiwanese students.

## 2. Materials and Methods

### 2.1. Dataset

Data were retrieved from the Taiwan Adolescent to Adult Longitudinal Study (TAALS) [32], a school-based nationally representative longitudinal survey with wave 1 in 2014 (baseline) and wave 2 in 2016 (follow-up). School was the primary sampling unit and the first wave included 18,064 first-year students from junior high school (*N* = 6667, 7th grade, mean age 13), senior high school (*N* = 4689, 10th grade, mean age 16), and vocational high school (*N* = 6708, 10th grade, mean age 16). The 15,795 students (87%) who completed the follow-up questionnaire two years later were used as the primary data set. The TAALS was approved by the Joint Institutional Review Board of Taipei Medical University, Taiwan (TMU-JIRB-201410043).

Table 1 shows the sample characteristics by ever-smoking status at follow-up. Among the 12,954 students who had never smoked at baseline, 1115 (8.6%) were ever smokers at 2-year follow-up. The attrition analysis additionally performed to assess whether baseline never smokers lost at follow-up were different from those retained with respect to the measures assessed at baseline indicates that in terms of age, gender, father’s education, and mother’s ethnicity the two groups are extremely similar (see Appendix A—Table A1).

### 2.2. E-Cigarette Use and Smoking Status at Baseline

At baseline, all participants were asked to answer “Yes” or “No” to the following questions: “Have you ever tried e-cigarettes?”, and “Have you ever smoked cigarettes, even one or two puffs?”.

### 2.3. Outcome Variable: Cigarette Smoking Initiation at Follow-Up

All students who responded “No” to having tried conventional cigarettes at baseline were asked again whether they had ever smoked at follow-up. Those who responded “Yes” were classified as having initiated smoking.

Due to the low number of established smokers in our dataset, it was not possible to take into account regular smoking (i.e., daily, weekly, or monthly). In Taiwan, smoking initiation occurs later than in Western countries [33]. Moreover, any smoking in adolescence predicts smoking in young adulthood [34].

### 2.4. Susceptibility to Smoking at Baseline

In case of cigarette-naïve individuals, susceptibility to smoking at baseline, defined as “absence of a firm decision not to smoke” [35], was assessed by asking “If one of your best friends offered you a cigarette would you smoke it?” and “At any time during the next 12 months do you think you will smoke a cigarette?” (answer options were “Definitely No”, “Probably No”, “Probably Yes”, and “Definitely Yes”). Participants who responded “Definitely No” to both questions were classified as not susceptible to smoking at baseline; all others were classified as susceptible.

### 2.5. Socio-Demographic Profile

The socio-demographic profile of each student was defined by age in years at the time of the survey, father’s education, mother’s ethnicity, parents’ employment status (full-time job, part-time job, or unemployed), and family living arrangement.

Only mother’s ethnicity was assessed because most Taiwanese newborns with one foreign parent are given birth by mothers of foreign origin (10% to 13% of the total number of newborns between 2001 and 2007 [36]), while very few newborns have fathers of foreign origin. Moreover, mothers from foreign countries may face different challenges (in terms of language and/or cultural and social networking) that may affect their children’s health behaviors, such as their willingness to try smoking. The education level of each parent is an important factor as well but due to limited questions allowed in the questionnaire, father’s education was chosen as representative of the educational level in the household.

### 2.6. Depression Status and Peer Support

The depression status of each participant was measured using the Center for Epidemiologic Studies Depression Scale (CES-D) [37]. The CES-D scale consists of 20 self-report items developed to measure depressive symptoms in the general population. The response choices for each item are “Not at all” (0 Points), “A little” (1 point), “Some” (2 points), and “A lot” (3 points). Items reflecting positive affect and behavior score reversely as 0 points for answering “A lot”, 1 point for “Some”, 2 points for “A little”, and 3 points for “Not at all”. The CES-D scale then ranges from 0 to 60. Peer support, i.e., the degree of support provided by friends (e.g., “During the past 6 months, were your friends there for you whenever you needed help?”), was quantified using a 5-item questionnaire developed by the U.S. Centers for Disease Control and Prevention, CDC [38]. Item responses were rated on a 4-point scale ranging from zero to three and were summed to calculate the total score. The index ranges from 0 to 15, and higher scores indicate stronger peer support.

### 2.7. Statistical Analysis

Data were weighted to be nationally representative of the Taiwanese student population. The following post-stratification weighting procedure was used. For each type of school (junior, senior, and vocational), we divided the sample into 24 groups, one for each combination of geographic area (North, South, Center, East), school size (small, medium, large), and gender. Then for each group we calculated a specific weight as follows:$$W_{g,s,x}=\left(N_{g,s,x}/N\right)/\left(n_{g,s,x}/n\right)$$
where $n_{g,s,x}$ is the number of individuals in the sample of gender x attending schools of size s located in the geographic area g and $N_{g,s,x}$ is the number of individuals in the population of gender x attending schools of size s located in the geographic area g. n is the size of the entire sample, and N is the size of the entire population (junior, senior, or vocational students). Further details can be found in a study by Chien et al. [32] We then applied complex survey logistic regression to determine whether students who had never smoked at baseline but had already used e-cigarettes were more likely to initiate smoking at 2-year follow-up than those who had not used e-cigarettes. Two models were considered: an unadjusted model just considering the crude relationship between ever use of e-cigarettes and smoking initiation, and an adjusted model including the following covariates: smoking susceptibility at baseline, socio-demographic profile, psychological status, and peer support. The interaction effect between school grade and ever use of e-cigarettes was initially analyzed and found to be not statistically significant (*p* = 0.93 for interaction). For this reason, the regression analyses were performed on the entire dataset.

A sensitivity analysis was also conducted to assess whether students lost to follow-up were different from those retained, and whether these differences might have influenced the results. In detail, we ran the above described regression analyses under two opposite scenarios: all the students lost at follow-up were non-smokers and all the students lost at follow-up were smokers. The Odds Ratios under the two scenarios were very similar (Appendix A—Table A2 and Table A3), leading us to conclude that respondents lost at follow-up may be different from those who retained, but this difference did not affect our main results significantly.

The statistical analysis was conducted using the “svyset” command in STATA statistical software (version 12, StataCorp LLC, College Station, TX, USA) and a *p* < 0.05 was considered significant.

## 3. Results

The unadjusted and adjusted logistic regressions provided similar results (Table 2), demonstrating that the effect of e-cigarette ever use on cigarette smoking initiation is independent of other determinants of smoking and that confounding is unlikely to be a problem. Students who had already tried e-cigarettes at baseline exhibited significantly higher odds of starting smoking in the following two years than those who never tried e-cigarettes; the odds ratios (ORs) being 2.44 (95% CI 1.94–3.09) with the unadjusted model and 2.14 (95% CI 1.66–2.75) with the adjusted model.

## 4. Discussion

In accordance with previous results in Western countries [7,23,24,39], our study shows that cigarette-naïve Taiwanese adolescents who had ever used e-cigarettes at baseline were more likely to initiate smoking 2 years later. In other words, starting to use e-cigarettes increased the odds of progressing to conventional cigarettes, which in Asian countries like Taiwan are still more affordable and ubiquitously available than e-cigarettes, which remain de facto illegal.

Smoking cessation is one reason that youth use e-cigarettes in Korea [3] and the USA [40]. In particular, Korean adolescents who tried to quit smoking are more likely to use e-cigarettes but less likely to no longer smoke, which suggests that e-cigarettes inhibit rather than promote cessation [3]. It has been argued that the observation that e-cigarettes increase the likelihood of subsequent cigarette smoking initiation merely reflects the fact that some users may share a similar propensity toward two behaviors [41]; however, our results seem to exclude the existence of other major confounders, as the odds ratios in our adjusted models were statistically significant.

The odds ratios found in our study are smaller than those reported in Western countries [14,16,17,20,23]. This difference could be due to the restrictive e-cigarette policy in Taiwan: the government has not yet approved any legal nicotine-based product and marketing of e-cigarettes is not allowed. In the current regulatory context, without visible marketing, frequent vaping scenes on TV, and exposure in daily life, adolescents may be less likely to use e-cigarettes.

The consistent finding of a prospective association between e-cigarette use and cigarette smoking in all the available longitudinal studies [7,15,23,24,39] suggests the need for policies to make e-cigarettes less attractive to youth. The European Union has recommended that a restrictive approach should be adopted towards advertising e-cigarettes and refill containers [26], while the availability of a wide range of refill flavors has been identified as an important motive for smoking e-cigarettes among young users [25], indicating that banning fruit-, mint-, and candy-flavored liquids would be a good starting point for prevention, as well as implementing the same restrictions on marketing currently applied to conventional cigarettes.

The results of Taiwan’s Global Youth Survey also revealed that smoking prevalence among youth has continued to decline, reaching historical lows in 2017 (8.3% among senior high school students, 2.7% among junior high school students), while at the same time e-cigarette use has increased [28]. Nicotine dependence may be the major reason pushing young “vapers” towards conventional smoking [39], especially given that cigarettes are largely affordable in Taiwan [42].

There are several strengths of this work. First, previous prospective studies on the association between e-cigarette use and subsequent smoking have all been from Western countries; our longitudinal study contributes to filling this gap for Asian youth [15]. Even though our findings come from a single medium-sized country, they are particularly important since other Asian countries share many similarities with Taiwan in terms of tobacco control, including former or current government-owned tobacco monopolies [43], high male smoking rate [44], lagged tobacco endemic transition, as well as tobacco control endeavors comparable to the Western countries. Second, the longitudinal design of the study ensures that e-cigarette use preceded initiation of cigarette smoking among never smoking adolescents. Third, our results are based on a national representative sample, with a low loss to follow-up. Fourth, the sensitivity analysis we performed to evaluate the possible impact of attrition revealed that even under two opposite scenarios (all the lost students were smokers and all the lost students were non-smokers) the ORs of initiating smoking were substantially identical: in the all-non-smokers case, 2.19 (*p* < 0.0001) with the unadjusted model and 2.03 (*p* < 0.0001) with the adjusted model; in the all-smokers case, 1.94 (*p* < 0.0001) with the unadjusted model and 2.03 (*p* < 0.0001) with the adjusted model. Fifth, no major changes in tobacco control occurred during the study period, hence external factors influencing smoking patterns remained mostly constant.

Nonetheless, there are also some limitations to this research. First, we were not able to determine if students referred to e-cigarettes containing nicotine or not, but about 80% of the 3062 e-cigarettes examined by Taiwan’s Food and Drug Administration in 2016 contained nicotine [45]. Second, while important covariates were included, rebellion, sensation seeking propensity, e-cigarette use and smoking by friends and parents, alcohol and other drugs use were not assessed in the survey. In particular, a recent study shows that sensation seeking is significantly related to experimentation with e-cigarettes, similarly to what happens with conventional cigarettes among adolescents [46], while Wills et al. (2016) found no significant association between prior use of e-cigarettes and subsequent initiation of smoking among youth after the inclusion of mediating variables such as marijuana use [47], contradicting previous results [11]. However, illicit substance use among adolescents in Taiwan is low (0.52% in 2014), with club drugs like N_2_O being the most popular [48]. In conclusion, other overlooked variables might explain the association between initial e-cigarette use and subsequent smoking.

One of the most plausible pathways for this sequential relationship is that adolescents who never smoked cigarettes are first exposed to addictive nicotine with e-cigarettes, thanks to sophisticated strategies of targeted marketing based on abundant attractive flavors and “less harmful” claims [49]; then, once nicotine use is established, adolescents become more open to conventional smoking.

## 5. Conclusions

We confirmed the prospective association of e-cigarette use with smoking initiation that has been observed in Western countries in an Asian population, with the largest effect among students exhibiting low susceptibility to initiating nicotine use with conventional cigarettes. With prevalence of e-cigarette smoking still relatively low in Taiwan, it is a good public health practice to maintain a restrictive and cautious policy towards e-cigarettes. Such measures are needed to prevent e-cigarettes from serving as a new gateway to conventional smoking for Asian youths, who are already exposed to a stronger smoking culture than their Western counterparts.

## Figures and Tables

**Table 1 ijerph-16-01145-t001:** Sample characteristics by ever smoking status at follow-up, among never smokers at baseline—unweighted data.

		Ever Smokers at Follow-Up (*N* = 1115)		Never Smokers at Follow-Up (*N* = 11,839)	
		*n*	% [*n*/*N*]	*n*	% [*n*/*N*]
Gender	Male	704	63.1%	4957	41.9%
	Female	4957	444.6%	6882	58.1%
Days of Smoking in the Past 30 Days	0	814	73.0%	11,837	100.0%
	1–5	156	14.0%	0	0.0%
	6–9	25	2.2%	0	0.0%
	10–19	29	2.6%	0	0.0%
	>20	90	8.1%	0	0.0%
School	Junior High	471	42.2%	5012	42.3%
	Senior High	644	57.8%	6827	57.7%
Ever Used E-Cigarettes at Baseline	Entire Dataset	118	10.6%	543	4.6%
	Junior High Students *	54	4.8%	253	2.1%
	Senior High Students †	64	5.7%	290	2.4%
Never Used E-Cigarettes at Baseline	Entire Dataset	209	18.7%	670	5.7%
	Junior High Students	96	8.6%	286	2.4%
	Senior High Students	113	10.1%	384	3.2%
Susceptible to Smoking at Baseline	Entire Dataset	209	18.7%	670	5.7%
	Junior High Students	96	8.6%	286	2.4%
	Senior High Students	113	10.1%	384	3.2%
Father’s Education	Below Junior High School	247	22.2%	2036	17.2%
	Senior or Vocational High School	452	40.5%	4396	37.1%
	Above College	295	26.5%	4298	36.3%
Mother’s Ethnicity	Native	939	84.2%	10,451	88.3%
	Indigenous	52	4.7%	337	2.8%
	Foreigner	106	9.5%	896	7.6%
Parents’ Employment Status	Full-time Job	1021	91.6%	11,093	93.7%
	Part-time Job	31	2.8%	211	1.8%
	Unemployed	51	4.6%	411	3.5%
Family Living Arrangement	Parents or Extended Family	794	71.2%	9433	79.7%
	Single Parents	242	21.7%	1886	15.9%
	Grandparents	33	3.0%	245	2.1%
	Other Relatives	46	4.1%	275	2.3%
		**Average**	**SD**	**Average**	**SD**
CES-D Scale		3.8	3.0	3.3	2.7
Peer Support Score		13.3	2.4	13.5	2.2

* 13 years old at baseline; † 16 years old at baseline. CES-D Scale: Center for Epidemiologic Studies Depression Scale.

**Table 2 ijerph-16-01145-t002:** Logistic regression output for smoking initiation—weighted estimates (observations: 11,615).

		Entire Dataset	
		ORs	95% CI
**Unadjusted Model**			
Ever Use of E-cigarettes at Baseline		2.44 ***	(1.94–3.09)
**Adjusted Model**			
Ever Use of E-cigarettes at Baseline		2.14 ***	(1.66–2.75)
Susceptibility to Smoking at Baseline		3.61 ***	(2.99–4.36)
Father’s Education	Below Junior High School	1.00	
	Senior or Vocational High School	0.91	(0.78–1.07)
	Above College	0.65 ***	(0.53–0.79)
Mother’s Ethnicity	Native	1.00	
	Indigenous	1.73 ***	(1.22–2.45)
	Foreigner	1.18	(0.93–1.49)
Parents’ Employment Status	Full-time Job	1.00	
	Part-time Job	1.29	(0.84–1.99)
	Unemployed	1.10	(0.78–1.57)
Family Living Arrangement	Parents or Extended Family	1.00	
	Single Parents	1.26 ***	(1.08–1.47)
	Grandparents	2.06 ***	(1.50–2.82)
	Other Relatives	1.53 *	(1.06–2.22)
Age (in years)		1.01	(0.96–1.07)
CES-D Scale		1.04 ***	(1.02–1.06)
Peer Support Score		0.98	(0.95–1.01)
Constant		0.07 ***	(0.02–0.21)

*** denotes *p* value < 0.001; ** denotes *p* value < 0.01; * denotes *p* value < 0.05. ORs: Odds Ratios.

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
