# Peer review of "Electronic Cigarette Use and Smoking Initiation in Taiwan: Evidence from the First Prospective Study in Asia"

_ijerph, 2019, doi:10.3390/ijerph16071145_

Round 1

Reviewer 1 Report

Summary

Employing data from a longitudinal survey of adolescents in Taiwan, this study examines the relationship between use of e-cigarettes among youth who never smoked cigarettes in 2014 and subsequent uptake of cigarette smoking two years later in 2016. The data for this study come from a nationally representative sample of 13,108 Taiwanese youth surveyed in both years. Youth who reported having ever used e-cigarettes in 2014 had significantly greater odds of having started to smoke by 2016, even after controlling for important confounding factors. This study is methodologically sound and well-written and represents one of the first studies examining the relationship between e-cigarette use and subsequent initiation of cigarette smoking in an Asian population. Having said that, I have some concerns that need to be addressed to further strengthen the study's findings.

Mandatory Revisions

1.  While "any" cigarette smoking initiation is a reasonable outcome to use for the analysis of smoking uptake in adolescents, I think additional information could be provided, at least in Table 1, about the proportion of youth ever smoking at follow-up who were smoking at least regularly (which, among youth, is typically taken to mean smoking at least weekly). Given the survey data, was it possible to measure more regular smoking with this sample? Would it be possible to consider experimental smoking (such as smokes < weekly) + regular smoking (smokes at least weekly) as the outcome for the analysis? Would findings change? While any initiation is important, from a gateway perspective, the bigger issue is whether youth who use e-cigarettes progress to more regular cigarette smoking. At the very least it is important to report the number and proportion of youth who started smoking who smoke at least monthly and at least weekly in Table 1.

2. It's also somewhat strange that youth who had used another type of combustible tobacco product but not cigarettes at baseline were retained in the analysis. Based on Table 1, there were only 154 of these youth in the sample at baseline, so I doubt removing them from the analysis would change the results for the regression models presented in Tables 2 and 3 in any meaningful way. However, including these youth in the analysis doesn't seem to be very clean to me, since they already have some experience with combustible tobacco. The analysis should be re-run excluding these youth to create a truly clean baseline sample of never smokers.

3. Page 4, section 2.6 – I think this section should provide more information about these two particular scales. For example, what is the range of the CES-D scale and do higher scores on this particular scale indicate greater levels of "depression"? Does this scale indicate some clinical form of depression or does it represent some other form of depression? What is the numerical range of the peer support scale? As a minor note, it's not clear that the items for the peer support scale are truly Likert style items, so the items could be more simply described as being rated "on a 4-point scale ranging from zero to three" (i.e., remove the word "Likert").

4. For the description of the calculation of the sampling weights, the authors state that weights were calibrated to national distributions for three variables: geographic area, school size and gender. However, the authors do not describe how this was done. Was a raking algorithm (a.k.a. iterative proportional fitting) employed? (This is often the case for post-calibration of sampling weights). Please clarify.

5. In section 2.7, the authors also state that data were analyzed using the entire data and also stratifying the analysis by dividing participants into junior and senior high school students. A few comments:

  Results presented in Tables 2 and 3 are stratified by baseline age (13 or 16). However, the logistic regression models also include "age" as a covariate, which seems to be age in years. This could be confusing, so results of the stratified analysis, in my mind, are better described as being stratified by grade or school year, i.e., junior vs. senior high school students.

  If this description represents what was actually done, then the "age" measure should be more clearly described in the methods section as reported age (in years) at the time of survey. As it stands, it's not clear what age in the models represents. For example, could "junior" students at baseline have been 12, 13, or 14 years of age? Is this what is meant when "age" is used as a covariate in the regression models? Please clarify.

  While I recognize that results may differ by age (or school grade in this case), it's not clear to me why the authors decided to stratify the analysis without testing a school grade by ever-used e-cigarettes at baseline interaction effect? Testing the interaction effect explicitly would provide justification for splitting the analysis by school grade. Given the similarity of the estimated odds ratios for most covariates, and for the main exposure variable (baseline use of e-cigarettes), the estimated OR for the overall analysis in Table 2 would seem to be a good summary measure of the odds of smoking initiation. The authors should explicitly test whether school grade modifies the relationship between baseline e-cigarette using and smoking initiation (i.e., test and report the interaction effect. If the interaction is significant, it would be more reasonable to present stratified results as the authors do. If the interaction is not significant, the authors could report the overall analysis only).

6. An interaction effect could also be estimated and tested for Table 3. Most importantly, the authors should test whether baseline susceptibility to smoking modified the relationship between baseline e-cigarette use and smoking initiation. As the results are currently presented (stratified by school grade and baseline susceptibility) it would seem as though the authors are interested in the three-way interaction between school grade (or age) X baseline susceptibility X baseline e-cigarette use. At the very least, the authors could test the two-way interaction first (between baseline susceptibility and baseline e-cigarette use) and report the results of that test. If significant, the authors might have some justification to stratify the results for table 3 by baseline susceptibility. (It's not clear whether stratifying by school grade is necessary).

 in addition, explicitly testing this two-way interaction effect would provide more convincing evidence for the argument made in the second paragraph of the Discussion section (lines 163–168 on page 7)

– moreover, if the results for baseline susceptibility are more important, I think Table 3 could be redesigned so that the columns of the table show results for the entire data set, then results for non-susceptible youth followed by results for susceptible youth. If necessary, the different sections of the table could show results for junior students (age 13) at baseline as the top half of the table followed by results for senior students (age 16) at the bottom half of the table.

7. Related to the stratification of results, since the data arise from a complex survey design, are the regression models the result of a "domain" analysis using the survey routines in Stata? Please clarify whether this is the case or whether subsets of data were created to estimate the regression models. The more correct way to analyse the data would be a domain analysis. But first, it would be important to test the interaction effects, as mentioned above. If significant, the authors should report this and estimate the stratified results using a domain analysis with the survey routines in Stata.

8. The titles for all tables should indicate whether the results presented represent weighted or unweighted estimates. The results in Table 1 seem to be unweighted estimates. Please make this clear in the title for Table 1. The results for Tables 2 and 3 are, I'm assuming, weighted estimates. This also needs to be explicitly stated. In addition, the sample sizes should be reported in Tables 2 and 3 as well (e.g., the "Entire Dataset column" should also report the model n as well as the next two columns. The same comment applies to Table 3).

9. In the first paragraph of the Discussion, the authors state that youth with low susceptibility to cigarette smoking have larger risks of having started to smoke at the 2 year follow-up (lines 157 to 160). However, it's not clear that this conclusion is warranted, given the authors did not explicitly test the interaction effect between baseline susceptibility and baseline e-cigarette use. While the estimated odds ratios for baseline e-cigarette use are smaller in magnitude for susceptible youth compared to non-susceptible youth, we can't tell on the basis of these results whether the estimates are statistically different. Testing the interaction effect would support this claim (if the overall interaction was significant, specific contrast tests should follow. In other words, using the entire data set, the authors could test whether the odds of starting to smoking in the non-susceptible group = odds in the susceptible group, or whether the OR = 2.31 in non-susceptible was significantly different from OR = 1.62 in the susceptible group).

10. The authors also claim in this paragraph that "once they start using e-cigarettes, many of these youth progress to conventional cigarettes" (lines 160-162) – the use of the word "many" here is a bit misleading in my mind. The overall prevalence of e-cigarette use is quite low (unweighted = 5.9%) and only 11.9% (or 135 youth according to Table 1) of baseline e-cigarette users had starting smoking 2 years later. I think this should described in terms of the relative odds of starting to smoking instead.

11. The authors correctly address the limitation of attrition bias in Appendix A. However, I'm not convinced that the presentation of ratios of relative frequencies (RRF) in Table A1 supports the authors claim that baseline never smokers lost at follow-up were similar to those retained (lines 69-74 on page 2). The RRFs are based largely on the same subset of respondents (i.e., the follow-up cohort makes up almost all of the baseline cohort). Therefore, it's no surprise that all RRFs are so close to 1. What's more important is whether those students lost to follow-up differ in meaningful ways from respondents who were retained and whether these differences might have influenced the results. So it's more instructive to compare the baseline differences of those lost to attrition against those who were retained/followed to Wave 2. For example, if respondents lost by wave 2 were more likely to use e-cigarettes at baseline than those who were retained, then this may have affected the association between baseline e-cigarette use and subsequent smoking initiation. It is this type of bias that needs to be assessed. This needs to be examined and explained clearly in the discussion. So baseline characteristics should be compared between those retained and those lost (e.g., the characteristics listed in Table A1 as well as baseline e-cigarette use and baseline susceptibility to smoking). If differences are found, might these differences affect your results, and if so, how?

Minor Comments

1. Page 5, line 125, change "smoking behavior regardless their prior…" to "smoking behavior regardless of their prior…"

2. Page 4, line 129, change "STATA statistic software" to "STATA statistical software".

3. In Table 2, not all of the significance stars are defined in the footnote. Since all the same stars are used as in Table 3, it seems the footnote to Table 3 also applies to Table 2. Please correct this.

Author Response

We truly appreciate all the constructive comments and suggestions from both reviewers. We have adopted most of the suggestions in our revised manuscript. The following are our point-to-point responses to the reviewers’ comments (the comments are shown with Italic and bold font).

REVIEWER 1

Open Review

(x)       I would not like to sign my review report

( )        I would like to sign my review report

English language and style

( )        Extensive editing of English language and style required

( )        Moderate English changes required

(x)       English language and style are fine/minor spell check required

( )        I don't feel qualified to judge about the English language and style

Yes

Can be improved

Must be improved

Not applicable

Does the introduction provide   sufficient background and include all relevant references?

(x)

( )

( )

( )

Is the research design   appropriate?

(x)

( )

( )

( )

Are the methods adequately   described?

(x)

( )

( )

( )

Are the results clearly presented?

( )

(x)

( )

( )

Are the conclusions supported by   the results?

(x)

( )

( )

( )

Comments and Suggestions for Authors

Summary

COMMENT: Employing data from a longitudinal survey of adolescents in Taiwan, this study examines the relationship between use of e-cigarettes among youth who never smoked cigarettes in 2014 and subsequent uptake of cigarette smoking two years later in 2016. The data for this study come from a nationally representative sample of 13,108 Taiwanese youth surveyed in both years. Youth who reported having ever used e-cigarettes in 2014 had significantly greater odds of having started to smoke by 2016, even after controlling for important confounding factors. This study is methodologically sound and well-written and represents one of the first studies examining the relationship between e-cigarette use and subsequent initiation of cigarette smoking in an Asian population.

RESPONSE: Thank you.

COMMENT: Having said that, I have some concerns that need to be addressed to further strengthen the study's findings.

RESPONSE: See specific responses below.

Mandatory Revisions

COMMENT: 1. While "any" cigarette smoking initiation is a reasonable outcome to use for the analysis of smoking uptake in adolescents, I think additional information could be provided, at least in Table 1, about the proportion of youth ever smoking at follow-up who were smoking at least regularly (which, among youth, is typically taken to mean smoking at least weekly).

RESPONSE: The information has been added to Table 1.

COMMENT: Given the survey data, was it possible to measure more regular smoking with this sample? Would it be possible to consider experimental smoking (such as smokes < weekly) + regular smoking (smokes at least weekly) as the outcome for the analysis? Would findings change? While any initiation is important, from a gateway perspective, the bigger issue is whether youth who use e-cigarettes progress to more regular cigarette smoking.

RESPONSE: Unfortunately, given the survey data, it was not possible to take into account regular smoking, due to the low number of established smokers. In Taiwan, smoking initiation occurs later than in Western countries. Moreover, any smoking in adolescence predicts smoking in young adulthood (Dutra L, Glantz S. Thirty-day smoking in adolescence is a strong predictor of smoking in young adulthood. Prev Med. 2018 Apr;109:17-21. doi: 10.1016/j.ypmed.2018.01.014. Epub 2018 Feb 3.). This point has been added to the manuscript, along with the corresponding reference.

COMMENT: At the very least it is important to report the number and proportion of youth who started smoking who smoke at least monthly and at least weekly in Table 1.

RESPONSE: The information has been added to Table 1.

COMMENT: 2. It's also somewhat strange that youth who had used another type of combustible tobacco product but not cigarettes at baseline were retained in the analysis. Based on Table 1, there were only 154 of these youth in the sample at baseline, so I doubt removing them from the analysis would change the results for the regression models presented in Tables 2 and 3 in any meaningful way. However, including these youth in the analysis doesn't seem to be very clean to me, since they already have some experience with combustible tobacco. The analysis should be re-run excluding these youth to create a truly clean baseline sample of never smokers.

RESPONSE: We redid the analysis as the reviewer suggested and revised the manuscript and tables accordingly. The reviewer is correct that the results did not materially change.

COMMENT: 3. Page 4, section 2.6 – I think this section should provide more information about these two particular scales. For example, what is the range of the CES-D scale and do higher scores on this particular scale indicate greater levels of "depression"? Does this scale indicate some clinical form of depression or does it represent some other form of depression? What is the numerical range of the peer support scale? As a minor note, it's not clear that the items for the peer support scale are truly Likert style items, so the items could be more simply described as being rated "on a 4-point scale ranging from zero to three" (i.e., remove the word "Likert").

RESPONSE: The CES-D scale consists of 20 self-report items developed to measure depressive symptoms in the general population. The response choices for each item are “Not at all” (0 Points), “A little” (1 point), “Some” (2 points), and “A lot” (3 points). Items reflecting positive affect and behavior score reversely as 0 points for answering “A lot”, 1 point for “Some”, 2 points for “A little”, and 3 points for “Not at all”. The CES-D scale then ranges from 0 to 60. Peer support, i.e. the degree of support provided by friends (e.g., “During the past 6 months, were your friends there for you whenever you need help?”), was quantified using a 5-item questionnaire developed by the U.S. CDC [38]. Item responses were rated on a 4-point Likert scale ranging from zero to three and were summed to calculate the total score. The index ranges from 0 to 15 and higher scores indicated stronger peer support. The manuscript has been revised to add this information.

COMMENT: 4. For the description of the calculation of the sampling weights, the authors state that weights were calibrated to national distributions for three variables: geographic area, school size and gender. However, the authors do not describe how this was done. Was a raking algorithm (a.k.a. iterative proportional fitting) employed? (This is often the case for post-calibration of sampling weights). Please clarify.

RESPONSE: The following post-stratification weighting procedure was used. For each type of school (junior, senior, and vocational), we divided the sample into 24 groups, one for each combination of geographic area (North, South, Center, East), school size (small, medium, large), and gender. Then for each group we calculated a specific weight as follows:

Where  is the number of individuals, in the sample, of gender  attending schools of size  located in the geographic area  and  is the number of individuals, in the population, of gender  attending schools of size  located in the geographic area .  is the size of the entire sample, and  is the size of the entire population (junior, senior, or vocational students).

Further details can be found in the supplementary material of Chien, Yu-Ning, Ping-Ling Chen, Yi-Hua Chen, Hsiu-Ju Chang, Suh-Ching Yang, Yi Chun Chen, and Hung-Yi Chiou. "The Taiwan Adolescent to Adult Longitudinal Study (TAALS): Methodology and Cohort Description." Asia Pacific Journal of Public Health 30, no. 2 (2018): 188-197.

The manuscript has been revised to add this information.

COMMENT: 5. In section 2.7, the authors also state that data were analyzed using the entire data and also stratifying the analysis by dividing participants into junior and senior high school students. A few comments:

RESPONSE: See specific responses below.

COMMENT: Results presented in Tables 2 and 3 are stratified by baseline age (13 or 16). However, the logistic regression models also include "age" as a covariate, which seems to be age in years. This could be confusing, so results of the stratified analysis, in my mind, are better described as being stratified by grade or school year, i.e., junior vs. senior high school students.

RESPONSE:  The Reviewer is right; age in years is one of the covariate while junior high students and senior high students are the two sub-sets. Due to its non-significant effect, the stratification by school grade was removed.

COMMENT: If this description represents what was actually done, then the "age" measure should be more clearly described in the methods section as reported age (in years) at the time of survey. As it stands, it's not clear what age in the models represents. For example, could "junior" students at baseline have been 12, 13, or 14 years of age? Is this what is meant when "age" is used as a covariate in the regression models? Please clarify.

RESPONSE: As already stated in the previous response, age represents the age in years at the time of the survey. At baseline, only first-year students were sampled, i.e. 7th grade in JHS (mean age 13 years old) and 10th grade in SHS (mean age 16 years old).

COMMENT: While I recognize that results may differ by age (or school grade in this case), it's not clear to me why the authors decided to stratify the analysis without testing a school grade by ever-used e-cigarettes at baseline interaction effect? Testing the interaction effect explicitly would provide justification for splitting the analysis by school grade. Given the similarity of the estimated odds ratios for most covariates, and for the main exposure variable (baseline use of e-cigarettes), the estimated OR for the overall analysis in Table 2 would seem to be a good summary measure of the odds of smoking initiation. The authors should explicitly test whether school grade modifies the relationship between baseline e-cigarette using and smoking initiation (i.e., test and report the interaction effect. If the interaction is significant, it would be more reasonable to present stratified results as the authors do. If the interaction is not significant, the authors could report the overall analysis only).

RESPONSE: We included an interaction analysis in Appendix B. As for the school grade, the interaction was not significant; consequently that stratification was dropped.

COMMENT: 6. An interaction effect could also be estimated and tested for Table 3. Most importantly, the authors should test whether baseline susceptibility to smoking modified the relationship between baseline e-cigarette use and smoking initiation. As the results are currently presented (stratified by school grade and baseline susceptibility) it would seem as though the authors are interested in the three-way interaction between school grade (or age) X baseline susceptibility X baseline e-cigarette use. At the very least, the authors could test the two-way interaction first (between baseline susceptibility and baseline e-cigarette use) and report the results of that test. If significant, the authors might have some justification to stratify the results for table 3 by baseline susceptibility. (It's not clear whether stratifying by school grade is necessary).

– in addition, explicitly testing this two-way interaction effect would provide more convincing evidence for the argument made in the second paragraph of the Discussion section (lines 163–168 on page 7)

– moreover, if the results for baseline susceptibility are more important, I think Table 3 could be redesigned so that the columns of the table show results for the entire data set, then results for non-susceptible youth followed by results for susceptible youth. If necessary, the different sections of the table could show results for junior students (age 13) at baseline as the top half of the table followed by results for senior students (age 16) at the bottom half of the table.

RESPONSE: Following the previous response, the interaction effect was estimated and it was found to be not significant. The results were modified accordingly.

COMMENT: 7. Related to the stratification of results, since the data arise from a complex survey design, are the regression models the result of a "domain" analysis using the survey routines in Stata? Please clarify whether this is the case or whether subsets of data were created to estimate the regression models. The more correct way to analyse the data would be a domain analysis. But first, it would be important to test the interaction effects, as mentioned above. If significant, the authors should report this and estimate the stratified results using a domain analysis with the survey routines in Stata.

RESPONSE: See previous response; as noted above the interaction was not significant and the analysis in question was dropped.

COMMENT: 8. The titles for all tables should indicate whether the results presented represent weighted or unweighted estimates. The results in Table 1 seem to be unweighted estimates. Please make this clear in the title for Table 1. The results for Tables 2 and 3 are, I'm assuming, weighted estimates. This also needs to be explicitly stated. In addition, the sample sizes should be reported in Tables 2 and 3 as well (e.g., the "Entire Dataset column" should also report the model n as well as the next two columns. The same comment applies to Table 3).

RESPONSE: The tables have been revised as suggested.

COMMENT: 9. In the first paragraph of the Discussion, the authors state that youth with low susceptibility to cigarette smoking have larger risks of having started to smoke at the 2 year follow-up (lines 157 to 160). However, it's not clear that this conclusion is warranted, given the authors did not explicitly test the interaction effect between baseline susceptibility and baseline e-cigarette use. While the estimated odds ratios for baseline e-cigarette use are smaller in magnitude for susceptible youth compared to non-susceptible youth, we can't tell on the basis of these results whether the estimates are statistically different. Testing the interaction effect would support this claim (if the overall interaction was significant, specific contrast tests should follow. In other words, using the entire data set, the authors could test whether the odds of starting to smoking in the non-susceptible group = odds in the susceptible group, or whether the OR = 2.31 in non-susceptible was significantly different from OR = 1.62 in the susceptible group).

RESPONSE: The Reviewer is right. Since the interaction effect is not significant, we cannot support this claim. We modified the manuscript accordingly.

COMMENT: 10. The authors also claim in this paragraph that "once they start using e-cigarettes, many of these youth progress to conventional cigarettes" (lines 160-162) – the use of the word "many" here is a bit misleading in my mind. The overall prevalence of e-cigarette use is quite low (unweighted = 5.9%) and only 11.9% (or 135 youth according to Table 1) of baseline e-cigarette users had starting smoking 2 years later. I think this should described in terms of the relative odds of starting to smoking instead.

RESPONSE: The manuscript has been revised as suggested.

COMMENT: 11. The authors correctly address the limitation of attrition bias in Appendix A. However, I'm not convinced that the presentation of ratios of relative frequencies (RRF) in Table A1 supports the authors claim that baseline never smokers lost at follow-up were similar to those retained (lines 69-74 on page 2). The RRFs are based largely on the same subset of respondents (i.e., the follow-up cohort makes up almost all of the baseline cohort). Therefore, it's no surprise that all RRFs are so close to 1. What's more important is whether those students lost to follow-up differ in meaningful ways from respondents who were retained and whether these differences might have influenced the results. So it's more instructive to compare the baseline differences of those lost to attrition against those who were retained/followed to Wave 2. For example, if respondents lost by wave 2 were more likely to use e-cigarettes at baseline than those who were retained, then this may have affected the association between baseline e-cigarette use and subsequent smoking initiation. It is this type of bias that needs to be assessed. This needs to be examined and explained clearly in the discussion. So baseline characteristics should be compared between those retained and those lost (e.g., the characteristics listed in Table A1 as well as baseline e-cigarette use and baseline susceptibility to smoking). If differences are found, might these differences affect your results, and if so, how?

RESPONSE: In the Appendix section we added a sensitivity analysis in which we assumed alternatively that all the students lost at follow-up were nonsmokers at follow-up and that all the students lost at follow-up were smokers at follow-up, in order to see if the results are different. In the unadjusted model, if they are all nonsmokers, the OR is 2.19 (P<0.0001); if they are all smokers, the OR is 1.94 (P<0.0001). In the adjusted model, if they are all nonsmokers, the OR is 2.03 (P<0.0001); if they are all smokers, the OR is 2.03 (P<0.0001). The ORs are identical in the adjusted model and only slightly different in the unadjusted model. We think respondents lost at follow-up may be different from those who retained but the difference did not affect our main results significantly.

We included this explanation at the end of the Statistical Analysis sub-section and we added the following sentence as a strength of our paper:

Fourth, the sensitivity analysis we performed to evaluate the possible impact of follow-up loss revealed that even under two opposite scenarios (all the lost students were smokers and all the lost students were non-smokers) the ORs of initiating smoking were substantially identical: in the all-non-smokers case 2.19 (P<0.0001) with the unadjusted model and 2.03 (P<0.0001) with the adjusted model; in the all-smokers case 1.94 (P<0.0001) with the unadjusted model and 2.03 (P<0.0001) with the adjusted model.

Minor Comments

COMMENT: 1. Page 5, line 125, change "smoking behavior regardless their prior…" to "smoking behavior regardless of their prior…"

RESPONSE: The manuscript has been revised as suggested.

COMMENT: 2. Page 4, line 129, change "STATA statistic software" to "STATA statistical software".

RESPONSE: The manuscript has been revised as suggested.

COMMENT: 3. In Table 2, not all of the significance stars are defined in the footnote. Since all the same stars are used as in Table 3, it seems the footnote to Table 3 also applies to Table 2. Please correct this.

RESPONSE: The table has been corrected as suggested.

REVIEWER 2

Open Review

(x)       I would not like to sign my review report

( )        I would like to sign my review report

English language and style

( )        Extensive editing of English language and style required

( )        Moderate English changes required

(x)       English language and style are fine/minor spell check required

( )        I don't feel qualified to judge about the English language and style

Yes

Can be improved

Must be improved

Not applicable

Does the introduction provide   sufficient background and include all relevant references?

(x)

( )

( )

( )

Is the research design   appropriate?

( )

( )

(x)

( )

Are the methods adequately   described?

(x)

( )

( )

( )

Are the results clearly presented?

(x)

( )

( )

( )

Are the conclusions supported by   the results?

( )

(x)

( )

( )

Comments and Suggestions for Authors

The authors report a longitudinal study in which Taiwanese teens were surveyed in 2014 and again in 2016, substantially improving on cross-section methodology. Subgroups had average ages of 13 and 16 in 2014. The analysis focuses on 13,108 individuals who were never smokers at baseline, amongst whom 1,137 became ever-smokers at follow-up.

COMMENT: While the analysis presented is probably technically correct, it could be enormously misleading from a public health or population risk management perspective. The conclusions are seriously overstated because the focus is too narrow.

RESPONSE: For the reasons we state below, we respectfully disagree.

COMMENT: The first sentence of the abstract overstates what is known about causality: “electronic cigarette use increases the risk of subsequent initiation of conventional smoking among cigarette-naïve adolescents.”

RESPONSE: The language used in the manuscript about “risk” is consistent with standard epidemiological usage.

COMMENT: While there is some association amongst these behaviors, it is plausible that recent declines in smoking prevalence among Western adolescents are due to competition from e-cigs. This is also consistent with the conclusions on lines 182-186.

RESPONSE: While this is a common argument among e-cigarette advocates, it is not supported by the evidence, particularly the consistent finding in the US that a substantial fraction of youth who initiate nicotine use with e-cigarettes have low susceptibility to smoking. Indeed, overall youth tobacco product use (cigarettes plus e-cigarettes) in the US has increased following the advent of e-cigarettes. See for instance Dutra, L.M.; Glantz, S.A. E-cigarettes and national adolescent cigarette use: 2004–2014. Pediatrics 2017, 139, e20162450; and Dutra, L.M.; Glantz, S.A. Thirty-day smoking in adolescence is a strong predictor of smoking in young adulthood. Prev. Med. 2018, 109, 17-21.

COMMENT: The analysis does not examine overall prevalence of smoking among teens over time, and it is not exactly possible to get this with the data described from TAALS. This is absolutely necessary to get a complete picture of the effect of e-cigs. For instance, this analysis does not count any benefit from teens smoking in 2014 who used e-cigs to quit by 2016 – they are not in the denominator.

RESPONSE: We are unaware of any paper demonstrating that e-cigarettes assist adolescent smokers in quitting and would appreciate the reviewer providing the citations so we can consider them. One cross-sectional study of Korean adolescents suggests that youth who use e-cigarettes are significantly less likely to be former smokers (Reference [3] in the revised manuscript).

COMMENT: The gateway hypothesis posits that probable non-smokers are seduced first by e-cigs, then advance to smoking. An alternative hypothesis is that potential smokers are unhappy or restless teens looking for solace, who try whatever behaviors might bring happiness and peace, in whatever order is convenient and practical. In this case the never smokers who tried e-cigs at age 13 have simply self-identified as unhappy or restless, and they are of course more likely to use some mood altering substance, compared to their happier, more tranquil peers.

RESPONSE: We controlled for these factors (as Depression Status and Peer Support) as did most of the other studies of the relationship between youth e-cigarette use and progression to cigarette smoking, all of which provide results inconsistent with the reviewer’s hypothesis.

COMMENT: It has been quite difficult to devise studies or analyses that distinguish between these alternatives. Smoking in the previous century must have frequently involved seduction of otherwise happy non-smokers, but the unhappy teen hypothesis resonates better with clinical experience over the last 20-30 years. The quite small effect of depression in the analysis makes me wonder if most of the depressed teens have already tried cigarettes, and so were excluded from the analysis.

RESPONSE: See previous responses.

COMMENT: The authors should think about these problems, perhaps add some information about the initial ever-smokers, and be more circumspect in conclusions. The longitudinal monitoring is good, but the picture is not complete. Even the longitudinal data set has been culled in a way that eliminates the chance of finding any benefit from e-cigs.

RESPONSE: See previous responses.

COMMENT: The idea that adolescents would “easily move” from vaping to smoking is only true if the regulatory environment makes vaping hard or costly to maintain. Vaping is much easier to hide, usually cheaper, less physically debilitating, easier to set aside for several hours, and nearly as satisfying compared to conventional cigarette smoking. Teens could like the somewhat faster and stronger hit from conventional cigarettes so much that they trade the other advantages of e-cigs to get that, but it is not a trivial trade-off to spend more time in withdrawal with a higher chance of detection from smelling awful and sneaking away to smoke.

RESPONSE: We removed the term “easily moved.”

COMMENT: Line 150, the inequality sign points the wrong direction, should be P<0.001.

RESPONSE: This typo has been corrected.

Reviewer 2 Report

The authors report a longitudinal study in which Taiwanese teens were surveyed in 2014 and again in 2016, substantially improving on cross-section methodology. Subgroups had average ages of 13 and 16 in 2014. The analysis focuses on 13,108 individuals who were never smokers at baseline, amongst whom 1,137 became ever-smokers at follow-up. 

While the analysis presented is probably technically correct, it could be enormously misleading from a public health or population risk management perspective. The conclusions are seriously overstated because the focus is too narrow.

The first sentence of the abstract overstates what is known about causality: “electronic cigarette use increases the risk of subsequent initiation of conventional smoking among cigarette-naïve adolescents.” While there is some association amongst these behaviors, it is plausible that recent declines in smoking prevalence among Western adolescents are due to competition from e-cigs. This is also consistent with the conclusions on lines 182-186.

The analysis does not examine overall prevalence of smoking among teens over time, and it is not exactly possible to get this with the data described from TAALS. This is absolutely necessary to get a complete picture of the effect of e-cigs. For instance, this analysis does not count any benefit from teens smoking in 2014 who used e-cigs to quit by 2016 – they are not in the denominator.

The gateway hypothesis posits that probable non-smokers are seduced first by e-cigs, then advance to smoking. An alternative hypothesis is that potential smokers are unhappy or restless teens looking for solace, who try whatever behaviors might bring happiness and peace, in whatever order is convenient and practical. In this case the never smokers who tried e-cigs at age 13 have simply self-identified as unhappy or restless, and they are of course more likely to use some mood altering substance, compared to their happier, more tranquil peers.

It has been quite difficult to devise studies or analyses that distinguish between these alternatives. Smoking in the previous century must have frequently involved seduction of otherwise happy non-smokers, but the unhappy teen hypothesis resonates better with clinical experience over the last 20-30 years. The quite small effect of depression in the analysis makes me wonder if most of the depressed teens have already tried cigarettes, and so were excluded from the analysis.

The authors should think about these problems, perhaps add some information about the initial ever-smokers, and be more circumspect in conclusions. The longitudinal monitoring is good, but the picture is not complete. Even the longitudinal data set has been culled in a way that eliminates the chance of finding any benefit from e-cigs.

The idea that adolescents would “easily move” from vaping to smoking is only true if the regulatory environment makes vaping hard or costly to maintain. Vaping is much easier to hide, usually cheaper, less physically debilitating, easier to set aside for several hours, and nearly as satisfying compared to conventional cigarette smoking. Teens could like the somewhat faster and stronger hit from conventional cigarettes so much that they trade the other advantages of e-cigs to get that, but it is not a trivial trade-off to spend more time in withdrawal with a higher chance of detection from smelling awful and sneaking away to smoke.

Line 150, the inequality sign points the wrong direction, should be P<0.001.

Author Response

(The authors gave the same response as above.)

Round 2

Reviewer 1 Report

The authors have addressed all of my initial comments and I think their responses strengthen the findings of their study. I have a few minor comments on this version of the draft that require minor changes to the paper.

Page 4 line 118: "Data were preliminary weighted to be nationally representative..." -- the "preliminary" qualifier is awkward in this context and not needed. Could simply be removed without changing the meaning of the sentence.

Similarly, in lines 134-136, the "preliminary" should be changed to something like "The interaction effect between school grade and ever use of e-cigarettes was initially analyzed and found to be not statistically significant (p = 0.93 for interaction)."

It looks like the supplementary tables (A2 and A3) in the appendix report the results of the adjusted model reported in Table 2 of the paper, rather than the results of the sensitivity analyses. Was this a copy/paste error? That is, all adjusted odds ratios in Tables A2 and A3 are the same as the adjusted odds ratios in Table 2. While the text reporting the results of the sensitivity analysis appears to be correct, please ensure that the estimates and test statistics in Tables A2 and A3 are correct.

It would also be beneficial to report the sample size the models in Tables A2 and A3 are based upon, since the additional observations lost to follow-up were included as non-smokers for the outcome (Table A2) and smokers (Table A3). 

New text page 6 (lines 193-198) - the initial part of the sentence seems a bit awkward -- maybe rephrase to "Fourth, the sensitivity analysis we performed to evaluate the possible impact of attrition revealed..."

Author Response

REVIEWER 1

Comments and Suggestions for Authors

The authors have addressed all of my initial comments and I think their responses strengthen the findings of their study. I have a few minor comments on this version of the draft that require minor changes to the paper.

COMMENT: Page 4 line 118: "Data were preliminary weighted to be nationally representative..." -- the "preliminary" qualifier is awkward in this context and not needed. Could simply be removed without changing the meaning of the sentence.

RESPONSE: Thank you for the suggestion. The word “preliminary” was removed.

COMMENT: Similarly, in lines 134-136, the "preliminary" should be changed to something like "The interaction effect between school grade and ever use of e-cigarettes was initially analyzed and found to be not statistically significant (p = 0.93 for interaction)."

RESPONSE: Thank you. We modified the sentence as suggested.

COMMENT: It looks like the supplementary tables (A2 and A3) in the appendix report the results of the adjusted model reported in Table 2 of the paper, rather than the results of the sensitivity analyses. Was this a copy/paste error? That is, all adjusted odds ratios in Tables A2 and A3 are the same as the adjusted odds ratios in Table 2. While the text reporting the results of the sensitivity analysis appears to be correct, please ensure that the estimates and test statistics in Tables A2 and A3 are correct.

RESPONSE: The Reviewer is right, thanks for noticing this big mistake. Data in Table A2 and A3 were corrected.

COMMENT: It would also be beneficial to report the sample size the models in Tables A2 and A3 are based upon, since the additional observations lost to follow-up were included as non-smokers for the outcome (Table A2) and smokers (Table A3).

RESPONSE: Thank you for the suggestion. We included the number of observations in Table A2 and A3 in the revised version.

COMMENT: New text page 6 (lines 193-198) - the initial part of the sentence seems a bit awkward -- maybe rephrase to "Fourth, the sensitivity analysis we performed to evaluate the possible impact of attrition revealed..."

RESPONSE: Thank you. We modified the sentence as suggested.

Thank you again for making this paper much improved. All authors appreciate your helps.  

Reviewer 2 Report

The authors have revised their report of a longitudinal study of Taiwanese teens

The authors request citations indicating that teens might use e-cigs for smoking cessation. As they state, much of the literature rejects this assertion, but like all things having to do with e-cigs, nuance is important. Here are some papers with positive conclusions.

E-cigarette initiation and associated changes in smoking cessation and reduction: the Population Assessment of Tobacco and Health Study, 2013–2015. DOI: 10.1136/tobaccocontrol-2017-054108

Current and Former Smokers’ Use of Electronic Cigarettes for Quitting Smoking: An Exploratory Study of Adolescents and Young Adults. DOIL 10.1093/ntr/ntw248

And regarding the hypothesis that e-cigs compete with tobacco cigarettes:

How does electronic cigarette access affect adolescent smoking? DOI: 10.1016/j.jhealeco.2015.10.003

This paper reports evidence of competition and simultaneously concludes that e-cigs are a gateway:

Electronic cigarette use and smoking initiation among youth: a longitudinal cohort study. DOI: 10.1503/cmaj.161002

Author Response

REVIEWER 2

Comments and Suggestions for Authors

The authors have revised their report of a longitudinal study of Taiwanese teens

The authors request citations indicating that teens might use e-cigs for smoking cessation. As they state, much of the literature rejects this assertion, but like all things having to do with e-cigs, nuance is important. Here are some papers with positive conclusions.

E-cigarette initiation and associated changes in smoking cessation and reduction: the Population Assessment of Tobacco and Health Study, 2013–2015. DOI: 10.1136/tobaccocontrol-2017-054108

RESPONSE: This paper is not relevant because it deals with adults.

Current and Former Smokers’ Use of Electronic Cigarettes for Quitting Smoking: An Exploratory Study of Adolescents and Young Adults. DOIL 10.1093/ntr/ntw248

RESPONSE: Thanks for pointing this out. This is reasonable to include.  It shows that cessation is an important reason kids use ecigs.  There are no data on actual efficacy for cessation. I added it to the paper.  Added at line 161 (reference 40).

And regarding the hypothesis that e-cigs compete with tobacco cigarettes: How does electronic cigarette access affect adolescent smoking? DOI: 10.1016/j.jhealeco.2015.10.003

RESPONSE: Thanks. This paper is an ecological analysis and was contradicted by a later paper that used individual level data (https://www.ncbi.nlm.nih.gov/pubmed/29422436).  

We don’t think it is worth including in our paper.

This paper reports evidence of competition and simultaneously concludes that e-cigs are a gateway: Electronic cigarette use and smoking initiation among youth: a longitudinal cohort study. DOI: 10.1503/cmaj.161002

RESPONSE: This is reasonable to include. We added it to the paper at line 188.

Thank you so much for helping our paper much improved. 
